# Egg Consumption in U.S. Children is Associated with Greater Daily Nutrient Intakes, including Protein, Lutein + Zeaxanthin, Choline, α-Linolenic Acid, and Docosahexanoic Acid

**DOI:** 10.3390/nu11051137

**Published:** 2019-05-22

**Authors:** Yanni Papanikolaou, Victor L. Fulgoni

**Affiliations:** 1Nutritional Strategies, Nutrition Research & Regulatory Affairs, 59 Marriott Place, Paris, ON N3L 0A3, Canada; 2Nutrition Impact, Nutrition Research, 9725 D Drive North, Battle Creek, MI 49014, USA; vic3rd@aol.com

**Keywords:** NHANES, children, adolescents, nutrients, eggs, diet quality, dietary patterns

## Abstract

Dietary pattern recommendations include consuming a variety of nutrient-dense foods in children and adolescents to promote optimal growth and development. The current study investigated associations with egg consumption and nutrient intakes, diet quality, and growth outcomes relative to non-egg consumers. The analysis used data from the U.S. National Health and Nutrition Examination Survey (NHANES) 2001-2012 in children and adolescents aged 2–18 years (*N* = 3,299, egg consumers; *N* = 17,030, egg non-consumers). Daily energy and nutrient intakes were adjusted for the complex sample design of NHANES using appropriate weights. Consuming eggs was associated with increased daily energy intake relative to non-egg consumption. Children and adolescents consuming eggs had elevated daily intake of protein, polyunsaturated, monounsaturated and total fat, α-linolenic acid, docosahexaenoic acid (DHA), choline, lutein + zeaxanthin, vitamin D, potassium, phosphorus, and selenium. Egg consumers had greater consumption, sodium, saturated fat, with reduced total and added sugar versus egg non-consumers. The analysis also showed that egg consumption was linked with lower intake of dietary folate, iron, and niacin. No associations were determined when examining diet quality and growth-related measures. A sub-analysis considering socioeconomic status showed that egg consumption was positively related with daily lutein + zeaxanthin and DHA intake. The current analysis demonstrated several nutrient-related benefits to support the continued inclusion of eggs in the dietary patterns of children and adolescents.

## 1. Introduction

Since the late 1960s, authoritative bodies have not recommended eggs as part of the diet, largely due to misperceptions resulting from insufficient data that egg consumption contributed to higher cholesterol levels and elevated risk of cardiovascular disease (CVD). In fact, in 1968, the American Heart Association published dietary recommendations to limit consumption of egg yolks to less than three/week [1]. A recent review has argued that this egg-specific dietary guidance was founded on mis-interpreted data that ultimately led to public health nutritional consequences [2]. With several countries removing dietary cholesterol restrictions from dietary guidance and questioning U.S dietary guidance [3,4,5,6], the 2015 Dietary Guidelines Advisory Committee (DGAC) [7] report reversed previous recommendations with the statement, “Previously, the Dietary Guidelines for Americans (DGA) recommended that cholesterol intake be limited to no more than 300 mg/day. The 2015 DGAC will not bring forward this recommendation because available evidence shows no appreciable relationship between consumption of dietary cholesterol and serum (blood) cholesterol, consistent with the American Heart Association/American College of Cardiology report. Cholesterol is not a nutrient of concern for overconsumption.”

The 2015–2020 DGA policy report [8] emphasizes increased consumption of vegetables, fruits, whole grains, low-fat/fat-free dairy products, and a variety of protein foods, including eggs, seafood, lean meats, legumes, nuts, and soy products. Further, dietary guidance has identified eggs to be nutrient-rich food products when consumed with minimal or no added sugars, sodium and/or solid fats. As such, all healthy dietary patterns recommended by the 2015–2020 DGA include eggs for Americans ≥2 years-old [8]. Nutrient-rich foods, like eggs, can contribute significantly to optimal childhood growth [9]. One 50 g egg (i.e., large egg) provides several essential nutrients and bioactives [10]. Particularly, eggs are an important dietary source of choline, an essential nutrient that is under consumed by the American population [11]. A 50 g egg contributes 146.9 mg of dietary choline and has been documented as a leading food source for choline in the American diet [10,12]. In addition, eggs contain omega-3 fatty acids, such that one large egg contributes about 30 mg docosahexaenoic acid (DHA) and 18 mg octadecatrienoic acid (α-linolenic-acid) [10]. A recent study in American children aged 7 to 12 years reported that few children meet recommendations for omega-3 fatty acids. In particular, only 6.8% of children had adequate intake of DHA + eicosapentaenoic acid (EPA) when using the United Nation Food and Agricultural Organization recommendations, while only 14.1% of children had adequate intakes when using the more lenient recommendations of the National Academy of Medicine [13]. A scientific EFSA panel has acknowledged that children’s brains accrue significant levels of DHA, with emphasis at infancy, but also throughout childhood, thus substantiating a cause and effect link for DHA dietary intake and neural physiology [14]. Eggs also provide highly bioavailable lutein and zeaxanthin, with a 50 g egg contributing approximately 250 mcg lutein + zeaxanathin [10]. Lutein and zeaxanthin are carotenoids which have been linked to eye health and reduced risk for eye- and vision-related diseases [15].

Previously published data in infants demonstrates several beneficial associations with egg consumption, of which included higher recumbent length compared to infants not consuming eggs. Furthermore, introducing eggs during infancy was linked to improved nutrient profiles, including higher daily intakes of protein, DHA, α-linolenic acid, phosphorus, choline, vitamin B12, and lutein + zeaxanthin [16]. At present, there are no studies in children and adolescents that have examined nutrient-related associations between consumers of eggs and non-egg consumers. As such, the purpose of the current investigation was to determine associations between egg consumers, nutrient intakes, diet quality, and growth-related outcomes in Americans 2-18 years of age, in comparison to egg non-consumers.

## 2. Materials and Methods

The present investigation used data from the US National Health and Nutrition Examination Survey (NHANES), which is a cross-sectional, nationally-representative, sample of U.S. free-living individuals. The data are compiled by the Centers for Disease Control and Prevention (CDC) in Atlanta, Georgia. Written informed consent and all ethical considerations have been previously approved by the appropriate ethics board at the CDC. For the current analysis, the methodology combined 6 datasets for individuals 2–18 years of age, beginning in 2001 and ending in 2012, to provide twelve years of data collection [17]. Data for nutrients are from the U.S. Department of Agriculture (USDA) Food and Nutrient Database for Dietary Studies (FNDDS) versions 5.0 and 6.0 for NHANES 2009–2010 and 2011–2012, respectively [18,19]. FNDDS serve as the databases which determine foods and beverages nutrient values in What We Eat in America (WWEIA) [19], which represents the dietary intake component of NHANES. WWEIA considers approximately 150 classifications, 15 central food groupings and 46 food subgroups. The collection procedure for WWEIA involves use of the Automated Multiple Pass Method (AMPM), representing a dietary collection tool that provides a valid, evidence-based approach for gathering data for national dietary surveys [20]. Although two days of recall are recorded in NHANES, the current analysis focused on 24-hour recalls obtained from Day 1 to represent data collected by an onsite interviewer [20,21]. Accuracy, effectiveness, and efficiency of the AMPM method has been extensively reported and previously documented [22,23,24].

### 2.1. Participants and Definition of Eggs

Male and female data on children and adolescents were combined for the present analysis and differentiated as consumers of eggs (*N* = 3,299) or egg non-consumers (*N* = 17,030). Only data that was determined to be reliable and included completed 24 hour recalled dietary data was included in the analysis. Egg consumption was defined strictly as participants that consumed only eggs, poached eggs, scrambled eggs, and omelets while excluding egg-containing mixed dishes (i.e., egg-containing sandwiches, breakfast burritos, and all egg-containing bakery foods, including cakes, breads, cookies, and biscuits).

### 2.2. Methodolgy

Statistical procedures were completed with the employment of SAS software (Version 9.2, SAS Institute, Cary, NC, USA) and SUDAAN 11.0. The investigation used survey weights to develop nationally representative estimates for children and adolescents, followed by adjustment to consider the complex sample design of the database. Adjusted means (± standard errors) for daily intake of energy (kilocalories), nutrients, and diet quality were determined. Energy, nutrient and diet quality included adjustment for several variables, including age, ethnicity, gender, kilocalories (i.e., all variables with the exception of energy intake), socioeconomic status (i.e., as measured by the poverty income ratio (PIR) and participation in the Special Supplemental Nutrition Program of Women, Infants and Children (WIC)) [24]. Similar adjustments were made for nutrient intakes, body weight, and growth measurements, with the inclusion of an adjustment for energy. USDA’s validated Healthy Eating Index-2010 (HEI 2010) tool was used to measure total diet quality—a measurement of alignment to authoritative dietary guidance [25].

## 3. Results

### 3.1. Population Demographics

Study population demographics can be viewed in Table 1. Consumers of eggs and non-egg consumers had differences in age, WIC and PIR status. For PIR, a greater value is representative of a larger income.

### 3.2. Daily Nutrient and Energy Intakes

Daily nutrient and energy intake comparisons for egg consumers and non-egg consumers can be seen in Table 2. Egg consumption was associated with higher protein, phosphorus, α-linolenic acid), DHA, polyunsaturated fat, monounsaturated fat, lutein + zeaxanthin, potassium, riboflavin, selenium, choline, vitamins D, A, and E. Egg consumption was associated with significantly lower daily added and total sugar intake. In contrast, consumers of eggs had reduced daily intakes of fiber, folate, and iron. Egg consumption was also linked to greater sodium, saturated and total fat intake compared to non-egg consumers.

### 3.3. Diet Quality Scores

The scores for total and sub-categories of Healthy Eating Index-2010 are presented in Table 3. The present analysis did not observe associations between the two egg groups when considering total diet quality, as assessed by HEI-2010. However, associations were apparent in certain sub-categories of HEI-2010. Specifically, consumption of eggs was associated with increased scores for fruits and vegetables, green beans, and total protein foods. Concurrently, egg consumption was also linked to decreased scores for whole grain and sodium consumption, implying that sodium is greater than recommended, and whole grain consumption is lower than recommended.

### 3.4. Sub-Analysis to Determine Intake Added Sugar, Carotenoids and Omega-3 Fatty Acids by Socio-Economic Status

In general, consumption of eggs in children and adolescents was linked to reduced daily intake for added sugar, but increased daily intake of lutein + zeaxanthin when compared to non-consumption of eggs in the defined age group. No differences were observed between daily intake of added sugar when subjects were classified as WIC participants. All egg consumers, regardless of their socioeconomic classification, had greater intake of docosahexaenoic acid relative to egg non-consumers. In all cases, except when children were classified as WIC participants, egg consumption was linked to higher α-linolenic acid daily intake when compared to non-egg consumption. In certain socioeconomic groups, but not all, egg consumers exhibited higher eicosapentaenoic acid intake compared to non-egg consumption. All results for added sugar, carotenoid and omega-3 fatty acid intake by socioeconomic status can be seen in Table 4.

### 3.5. Weight and Growth Measures

Table 5 provides results for weight and growth outcomes assessed. No significant differences were seen in weight and growth measures examined between egg consumers and egg non-consumers.

## 4. Discussion

The current NHANES analysis revealed significant associations with egg consumption in children and adolescents. A dietary pattern that includes eggs was linked with higher amounts of several nutrients, including protein, polyunsaturated and monounsaturated fat, α-linolenic acid, DHA, lutein + zeaxanthin, potassium, phosphorus, choline, riboflavin, selenium, choline, vitamins D, E, and vitamin A. Likewise, egg consumers had lower daily sugar intake (i.e., added and total sugar) when compared to children and adolescents not consuming eggs. Egg consumption was further associated with significantly lower daily intakes of dietary fiber, iron, and folate. The current study also showed that egg consumption was positively related with sodium intake, as well as saturated and total fat intake; thus, preparation of eggs, or foods that accompany eggs may require further investigation. In additional analyses considering socioeconomic status, the current data show benefits linked with egg consumption in this population. In general, egg consumption in children and adolescents, irrespective of food security, poverty and/or WIC status, was related with higher daily intake of lutein + zeaxanthin and DHA, and in most cases, reduced daily intake of added sugar versus the avoidance of eggs in the diet.

The present study further illustrates that choline intake is elevated in children and adolescents consuming eggs. Previous literature has documented the dietary importance of choline, largely due to choline’s relevance in physiology and metabolic activity [9,10,11,12] with several publications targeting the critical role choline plays in neuronal structures in early life [9,10,11]. While data shows that small amounts of choline can be generated endogenously, levels are not sufficient to meet physiological needs [26,27]. Most children consume less than the Adequate Intake (i.e., 550 mg for individuals greater than 4 years of age) [27,28]. A recent NHANES study showed that average choline intake in children and adolescents was approximately 256 mg per day [29]. Eggs have been identified as a leading dietary source of choline—a 50 g hard-boiled egg contributing 146.9 mg total choline or 27% of the recommended Daily Value [10]. As an example, the DGA Healthy Mediterranean-Style eating pattern recommends 5.5 oz equivalents for protein foods (not including dairy foods) within 1600 kcal daily [8]. If allowing for other protein-rich foods, including seafood, meat, poultry, nuts/seeds, etc., and adding two large 50 g eggs (2 oz equivalents) daily to the dietary pattern, a child’s daily choline intake would be approximately 295 mg, representing over 50% of the Adequate Intake for choline [10]. Additionally, since a large egg contributes a good source of dietary DHA [10], consuming 2 eggs daily would represent a DHA intake meeting 24% of EFSA’s daily recommendation for individuals 2-18 years-old (250 mg DHA) [14]. An EFSA expert panel noted a cause and effect mechanism in early brain and nervous system development during infancy and childhood with DHA intake resulting in the approval of claims for food marketing purposes [14].

Our previously published data also verified that consumption of eggs in infants and toddlers was related with increased daily nutrient intake relative to infants consuming no eggs in their dietary pattern [16]. Indeed, infants consuming egg had significantly higher daily intake of protein, α-linolenic acid, DHA, phosphorus, selenium, choline, lutein + zeaxanthin, and vitamin B12. Infants consuming eggs also had significantly reduced added and total sugar intake compared to non-consumers. However, infant egg consumption was also associated with lowered daily intakes of vitamins D, A, and E, in addition to reduced daily intake of three shortfall nutrients [7,8], including iron, potassium and dietary fiber. This may imply that other food groups, including grain foods, dairy foods, fruits and vegetables, may be key additions to early life eating patterns. Like the current data, the infant/toddler study showed that egg consumption was associated with higher saturated fat and sodium intake. While not investigated in the current study, foods that traditionally accompany eggs, including bacon and sausages, may be contributing to the increased sodium intake in the dietary pattern. This may imply the cooking methods involved with preparing and serving eggs and higher sodium foods that accompany eggs, may require additional scientific evaluation. Further investigation in this area is recommended, particularly with the release of a new report in older adults that showed higher consumption of cholesterol or eggs was associated with a small increase in CVD and all-cause mortality [30]. However, the study did not consider foods often consumed with eggs, including higher added sugar-, saturated fat-, and sodium-containing foods, like bacon, sausages, pancakes, waffles, and syrups.

Also aligned to the infant and toddler data, the current analysis in children and adolescents showed that eggs, as part of a dietary pattern, was not related with HEI, a measure of diet quality. Nevertheless, consumption of eggs in children and adolescents was associated with several HEI sub-categories, of which included, higher values for total fruits and vegetables, beans, and protein foods, but reduced scores for whole grains and total dairy consumption. Sodium scores were reflective of greater sodium intake in children and adolescents consuming eggs, suggesting that preparation of eggs and/or the types of foods that accompany egg meals (i.e., omelets with high-sodium foods including bacon) may need further investigation.

Further, in our current analysis, we did not observe associations with egg or non-egg consumption in several growth outcomes, including overweight, obesity, body weight, standing height or recumbent length. As previously discussed, while protein intake was elevated in children and adolescents consuming eggs, growth and development are multifactorial, thus numerous variables within a dietary pattern can impact such health outcomes [16].

Our analysis involving socioeconomic status revealed associations, such that greater daily lutein + zeaxanthin intake was linked to egg consumption, thus emphasizing the critical dietary role eggs can play with eye health in this population. In most cases, egg consumption was associated with reduced daily intake of added sugar relative to non-egg consumers, suggesting that increased egg consumption in children and adolescents may serve to help this population reduce added sugars in the diet through food selection, thus aligning with DGA recommendations. All egg consumers, regardless of their socioeconomic classification, had greater intake of docosahexaenoic acid relative to egg non-consumers. Additionally, except when children were classified as WIC participants, egg consumption was associated with higher intake of α-linolenic acid. Previous work has discussed how diet quality disproportions may potentially be linked to the higher costs of healthier dietary patterns [31], thus, from an economical perspective, eggs may offer nutrient density at a reasonable cost [32].

As has been documented previously in similar research methods [16], the current study has limitations inherent in observational research. NHANES provides a unique tool to researchers in that it offers a large cross-sectional database that pools together sophisticated, in-person interviews with validated physical and biochemical examinations. Limitations include memory recall bias with the 24-hour dietary recall; however, procedures are in place to reduce and minimize error introduction into the dataset. Further strengths and limitations have been previously published and discussed [33,34,35].

## 5. Conclusions

This data represents the first study in US children and adolescents to demonstrate nutrient intake associations when comparing egg consumption to non-consumption within a dietary pattern. Egg intake in this population was associated with increased daily intake of protein, polyunsaturated and monounsaturated fat, α-linolenic acid, DHA, lutein + zeaxanthin, choline, potassium, phosphorus, selenium, riboflavin, vitamins D, A, and E. Likewise, egg consumers had lower daily sugar intake (i.e., added/total sugar) when compared to children and adolescents not consuming eggs. Several shortfall nutrients were associated with egg consumption, including reduced daily intakes of dietary fiber, iron, and folate, concurrently with greater daily intake of sodium, total and saturated fat, suggesting that future research may need to evaluate the contribution of mixed egg meals and the type of foods accompanying eggs in the diet. In additional analyses considering socioeconomic status, the current data show benefits linked with egg consumption in this population. In general, egg consumption in children and adolescents, irrespective of food security, poverty income ratio and/or WIC status, was linked with elevated daily lutein + zeaxanthin and DHA intake, and in most cases, reduced added sugar intake. The present study further illustrates an opportunity to communicate the benefits linked with egg consumption to individuals that influence children and adolescents, including parents, school nutrition organizations, educators, and dietary guideline advisory committees globally.

## Figures and Tables

**Table 1 nutrients-11-01137-t001:** Mean variables for demographics when comparing egg non-consumers to egg consumers.

Variable	Egg Non-Consumers Sample *N* = 17,030 Population *N* = 59,475,530		Egg Consumers Sample *N* = 3299 Population N = 11,503,648		*p*
	Mean	SE	Mean	SE	
Age (Years)	10.16	0.07	9.38	0.17	<0.0001
Gender, Male (%)	50.76	0.64	50.00	1.34	0.6088
PIR < 1.35 (%)	31.98	1.04	39.24	1.78	0.0004
1.35 ≤ PIR ≤ 1.85 (%)	11.58	0.56	10.19	0.89	0.1862
PIR > 1.85 (%)	56.44	1.18	50.57	1.96	0.0103
WIC Participant (%)	14.25	0.62	18.37	1.13	0.0014
Full Food Security (%)	71.67	0.89	68.79	1.57	0.1097

Mean = least square mean; SE = standard error; PIR = Poverty Income Ratio; WIC = Special Supplemental Nutrition Program of Women, Infants and Children.

**Table 2 nutrients-11-01137-t002:** Day 1 nutrient and energy intakes in egg consumers vs. egg non-consumers.

Energy/Nutrients	Egg Non-Consumers		Egg Consumers		Beta	SE	*p*
	Mean	SE	Mean	SE			
Energy (kcal)	1959	10	2152	29	194	32	<0.0001
Carbohydrate (g)	270.9	0.7	248.0	1.6	−22.8	1.7	<0.0001
Added sugars (tsp eq)	21.1	0.2	18.2	0.4	−2.9	0.4	<0.0001
Total sugars (g)	137.5	0.7	124.7	1.7	−12.8	1.7	<0.0001
Protein (g)	68.4	0.3	75.5	0.7	7.1	0.8	<0.0001
Total fat (g)	72.1	0.2	78.4	0.6	6.3	0.7	<0.0001
Total MUFA (g)	26.2	0.1	28.6	0.3	2.4	0.3	<0.0001
Total PUFA (g)	14.5	0.1	15.5	0.2	1.0	0.2	<0.0001
Total SFA (g)	25.4	0.1	27.3	0.3	1.9	0.3	<0.0001
PUFA 18:3 (Octadecatrienoic) (g)	1.21	0.01	1.29	0.02	0.08	0.02	0.001
PUFA 20:5 (Eicosapentaenoic) (g)	0.01	0.0006	0.02	0.001	0.002	0.002	0.1194
PUFA 22:6 (Docosahexaenoic) (g)	0.03	0.001	0.06	0.002	0.035	0.002	<0.0001
Cholesterol (mg)	176.5	1.0	494.6	6.8	318.1	7.1	<0.0001
Dietary fiber (g)	13.2	0.1	12.4	0.2	−0.8	0.2	<0.0001
Calcium (mg)	1022.0	7.4	997.3	15.7	−24.7	16.7	0.1426
Folate, DFE (µg)	541.2	5.1	459.6	8.1	−81.6	10.1	<0.0001
Iron (mg)	14.5	0.1	13.6	0.2	−0.9	0.2	0.0002
Lutein + zeaxanthin (µg)	711.0	18.8	1035.5	49.2	324.5	53.6	<0.0001
Magnesium (mg)	230.9	1.2	229.0	2.4	−2.0	2.5	0.4269
Niacin (mg)	21.1	0.1	19.1	0.3	−2.1	0.3	<0.0001
Phosphorus (mg)	1247.7	5.3	1328.8	13.2	81.1	13.4	<0.0001
Potassium (mg)	2215.3	13.9	2274.5	24.1	59.2	26.1	0.0257
Riboflavin (Vitamin B2) (mg)	2.07	0.01	2.19	0.03	0.13	0.03	0.0001
Selenium (µg)	89.7	0.4	110.5	1.2	20.7	1.2	<0.0001
Sodium (mg)	3102.9	14.0	3305.3	28.2	202.4	29.7	<0.0001
Thiamin (Vitamin B1) (mg)	1.57	0.01	1.45	0.02	−0.12	0.02	<0.0001
Total choline (mg)	225.9	1.5	402.4	5.1	176.4	5.5	<0.0001
Vitamin A, RAE (µg)	575.6	6.4	640.4	14.5	64.8	16.0	0.0001
Vitamin B12 (µg)	5.02	0.05	5.26	0.12	0.24	0.14	0.0951
Vitamin B6 (mg)	1.70	0.01	1.65	0.03	−0.04	0.03	0.1583
Vitamin C (mg)	82.4	1.3	83.3	2.0	0.9	2.4	0.7046
Vitamin D (µg)	5.8	0.1	6.7	0.1	0.9	0.1	<0.0001
Vitamin E (mg)	5.9	0.1	6.6	0.2	0.7	0.2	0.0001
Zinc (mg)	10.6	0.1	10.3	0.1	−0.4	0.1	0.0079

LSMean = least square mean; SE = standard error; Beta = regression coefficient for difference between egg consumers vs. non-egg consumers; MUFA = monounsaturated fatty acids; PUFA = polyunsaturated fatty acids; SFA = saturated fatty acids; vitamin D = D_2_ and D_3_; vitamin E = as α-tocopherol.

**Table 3 nutrients-11-01137-t003:** Day 1 Healthy Eating Index-2010 (HEI) and Sub-Category Mean Scores.

HEI Total & 12 HEI Sub-Categories	Egg Non-Consumers		Egg Consumers		*p*
	Mean	SE	Mean	SE	
Total Vegetables (Category 1)	2.08	0.02	2.18	0.05	0.0358
Greens and Beans (Category 2)	0.57	0.02	0.74	0.05	0.0034
Total Fruit (Category 3)	2.47	0.04	2.63	0.06	0.0103
Whole Fruit (Category 4)	2.12	0.04	2.14	0.07	0.7418
Whole Grains (Category 5)	1.94	0.04	1.65	0.09	0.0030
Total Dairy (Category 6)	7.06	0.05	6.53	0.11	<0.0001
Total Protein Foods (Category 7)	3.41	0.02	4.26	0.03	<0.0001
Seafood and Plant Protein (Category 8)	1.43	0.03	1.28	0.08	0.0473
Fatty Acid Ratio (Category 9)	3.79	0.04	3.75	0.11	0.7088
Sodium (Category 10)	5.16	0.05	4.32	0.10	<0.0001
Refined Grains (Category 11)	5.18	0.05	6.00	0.11	<0.0001
SOFAAS Calories (Category 12)	10.18	0.10	10.72	0.21	0.0134
Total	45.39	0.23	46.21	0.44	0.0656

Mean = least square mean; SE = standard error; Beta = regression coefficient for difference between egg consumers vs. non-egg consumers; SOFAAS = solid fats, alcohol, added sugars.

**Table 4 nutrients-11-01137-t004:** Day 1 daily Intakes of added sugar, lutein/zeaxanthin, and omega-3 fatty acids by food security, poverty income ratio (PIR) and Special Supplemental Nutrition Program for Women, Infants, and Children (WIC) status.

Population	Nutrient	Egg Non-Consumers			Egg Consumers		SE	*p*
		*N*	Mean	SE	*N*	Mean		
Food Security = High	Added sugars (tsp eq)	10,353	21.1	0.2	1914	17.9	0.4	<0.0001
Food Security = High	Lutein + zeaxanthin (mcg)	10,353	709.2	23.3	1914	1073.0	64.7	<0.0001
Food Security = High	PFA 18:3 (Octadecatrienoic) (gm)	10,352	1.20	0.01	1915	1.28	0.03	0.025
Food Security = High	PFA 20:5 (Eicosapentaenoic) (gm)	10,352	0.01	0.0008	1915	0.02	0.001	0.034
Food Security = High	PFA 22:6 (Docosahexaenoic) (gm)	10,352	0.03	0.001	1915	0.06	0.003	<0.0001
Food Security = Low	Added sugars (tsp eq)	6261	21.0	0.3	1315	18.9	0.6	0.0016
Food Security = Low	Lutein + zeaxanthin (mcg)	6261	714.9	28.0	1315	946.6	50.8	<0.0001
Food Security = Low	PFA 18:3 (Octadecatrienoic) (gm)	6261	1.23	0.01	1315	1.33	0.03	0.004
Food Security = Low	PFA 20:5 (Eicosapentaenoic) (gm)	6261	0.02	0.001	1315	0.01	0.003	0.9327
Food Security = Low	PFA 22:6 (Docosahexaenoic) (gm)	6261	0.03	0.002	1315	0.06	0.003	<0.0001
PIR < 1.35	Added sugars (tsp eq)	7079	20.3	0.3	1582	18.9	0.5	0.0223
PIR < 1.35	Lutein + zeaxanthin (mcg)	7079	726.3	24.9	1582	937.4	47.1	<0.0001
PIR < 1.35	PFA 18:3 (Octadecatrienoic) (gm)	7079	1.23	0.01	1582	1.30	0.03	0.019
PIR < 1.35	PFA 20:5 (Eicosapentaenoic) (gm)	7079	0.02	0.001	1582	0.02	0.002	0.9048
PIR < 1.35	PFA 22:6 (Docosahexaenoic) (gm)	7079	0.03	0.002	1582	0.06	0.003	<0.0001
PIR ≥ 1.35	Added sugars (tsp eq)	8930	21.5	0.2	1504	17.7	0.5	<0.0001
PIR ≥ 1.35	Lutein + zeaxanthin (mcg)	8930	703.3	23.9	1504	1102.8	68.8	<0.0001
PIR ≥ 1.35	PFA 18:3 (Octadecatrienoic) (gm)	8929	1.20	0.01	1505	1.29	0.03	0.012
PIR ≥ 1.35	PFA 20:5 (Eicosapentaenoic) (gm)	8929	0.01	0.001	1505	0.02	0.002	0.045
PIR ≥ 1.35	PFA 22:6 (Docosahexaenoic) (gm)	8929	0.03	0.001	1505	0.06	0.003	<0.0001
WIC Participant = No	Added sugars (tsp eq)	10,815	21.2	0.2	1915	18.1	0.5	<0.0001
WIC Participant = No	Lutein + zeaxanthin (mcg)	10,815	719.7	22.5	1915	1078.6	67.3	<0.0001
WIC Participant = No	PFA 18:3 (Octadecatrienoic) (gm)	10,814	1.22	0.01	1916	1.32	0.03	0.001
WIC Participant = No	PFA 20:5 (Eicosapentaenoic) (gm)	10,814	0.01	0.001	1916	0.02	0.001	0.3449
WIC Participant = No	PFA 22:6 (Docosahexaenoic) (gm)	10,814	0.03	0.001	1916	0.06	0.003	<0.0001
WIC Participant = Yes	Added sugars (tsp eq)	2832	17.6	0.3	732	16.9	0.5	0.2954
WIC Participant = Yes	Lutein + zeaxanthin (mcg)	2832	630.1	21.1	732	852.5	29.9	<0.0001
WIC Participant = Yes	PFA 18:3 (Octadecatrienoic) (gm)	2832	1.17	0.03	732	1.22	0.03	0.1510
WIC Participant = Yes	PFA 20:5 (Eicosapentaenoic) (gm)	2832	0.01	0.002	732	0.01	0.002	0.2558
WIC Participant = Yes	PFA 22:6 (Docosahexaenoic) (gm)	2832	0.02	0.002	732	0.05	0.003	<0.0001

Mean = least square mean; SE = standard error.

**Table 5 nutrients-11-01137-t005:** Adjusted mean (SE) weight and growth measures for egg consumers vs. egg non-consumers.

Weight/Growth Measures	Egg Non-Consumers			Egg Consumers			*p*
Growth Variable	*N*	LSMean	SE	*N*	LSMean	SE	
Overweight	2,771	0.14	0.01	707	0.13	0.02	0.3671
Obese	2,771	0.18	0.01	707	0.22	0.02	0.1270
Overweight or Obese	2,771	0.32	0.01	707	0.35	0.03	0.4321
Body Weight (kg)	2,806	32.5	0.3	723	33.5	0.7	0.1276
Standing Height (cm)	2,769	124.8	0.2	706	124.0	0.5	0.1706
Recumbent Length (cm)	2,033	96.1	0.1	602	95.6	0.3	0.2363

LSMean = least square mean; SE = standard error; Data were gender combined; NHANES 2001–2012; Day 1 intakes; Covariates include age, gender, ethnicity, poverty income ratio, and energy intake (kcal).

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
