# Peer review of "Egg Consumption in U.S. Children is Associated with Greater Daily Nutrient Intakes, including Protein, Lutein + Zeaxanthin, Choline, α-Linolenic Acid, and Docosahexanoic Acid"

_nutrients, 2019, doi:10.3390/nu11051137_

Round 1

Reviewer 1 Report

This paper describes differences in nutrient profiles in participants who consume eggs versus those who do not. The research is novel and relevant to the field. The manuscript follows the correct structure and is well written.

The abstract appropriately summarizes the results and key findings.

The introduction covers the relevant background and perspective for this study. This paper nicely summarizes the studies considering the general risks/benefits of egg consumption that have changed over the decades, however, one new study is missed in the introduction - a recent study came out last month, that again speaks to the association with egg consumption and heart disease (see: Zhong VW, Van Horn L, Cornelis MC, et al. Associations of Dietary Cholesterol or Egg Consumption With Incident Cardiovascular Disease and Mortality. JAMA. 2019;321(11):1081–1095. doi:10.1001/jama.2019.1572, and NPR wrote up a news story, see: https://www.npr.org/sections/thesalt/2019/03/16/703804493/egg-lovers-new-study-finds-eating-too-many-can-increase-risk-of-heart-disease).

The methodology seems sound and reproducible. 

The results section clearly presents the results from this study, in a straightforward and easy to follow format. 

The discussion nicely addresses the results in the context of other studies. There is mention of egg consumers taking in more sodium and fat, and less sugar, which may be because bacon or sausage are often consumed with eggs, and by eating eggs, are eating less sugary breakfast items (like pancakes/waffles or cereal.) Some additional speculation/description towards this end may be nice to include.

Author Response

Dear Reviewer,

Thank you for taking the time to review our paper and provide feedback. We have provided answers and comments to your questions and suggestions below. Please let us know if you have any further questions or suggestions.

Sincerely,

Yanni Papanikolaou and Victor Fulgoni

Reviewer’s Question: The discussion nicely addresses the results in the context of other studies. There is mention of egg consumers taking in more sodium and fat, and less sugar, which may be because bacon or sausage are often consumed with eggs, and by eating eggs, are eating less sugary breakfast items (like pancake/waffles or cereals). Some additional speculation/description towards this end may be nice to include.

Author’s Response: Thank you for providing this comment. We agree with your recommendation and have added the following in the Discussion section on pages 9 and 10:

“While not investigated in the current study, foods that traditionally accompany eggs, including bacon and sausages, may be contributing to the increased sodium intake in the dietary pattern. This may imply the cooking methods involved with preparing and serving eggs and higher sodium foods that accompany eggs, may require additional scientific evaluation. Further investigation in this area is recommended, particularly with the release of a new report in older adults that showed higher consumption of cholesterol or eggs was associated with a small increase in CVD and all-cause mortality [30]. However, the study did not consider foods often consumed with eggs, including higher added sugar-, saturated fat-, and sodium-containing foods, like bacon, sausages, pancakes, waffles, and syrups.”

Reviewer’s Suggestion: The introduction covers the relevant background and perspective for this study. This paper nicely summarizes the studies considering the general risks/benefits of egg consumption that have changed over the decades, however, one new study is missed in the introduction - a recent study came out last month, that again speaks to the association with egg consumption and heart disease (see: Zhong VW, Van Horn L, Cornelis MC, et al. Associations of Dietary Cholesterol or Egg Consumption With Incident Cardiovascular Disease and Mortality. JAMA. 2019;321(11):1081–1095. doi:10.1001/jama.2019.1572, and NPR wrote up a news story, see: https://www.npr.org/sections/thesalt/2019/03/16/703804493/egg-lovers-new-study-finds-eating-too-many-can-increase-risk-of-heart-disease).

Author’s Response: Thank you for providing this recommendation. While we were aware of this paper published days before submission of our article, we decided not to include in our paper, since the JAMA paper focuses on adults and not children or adolescents. However, it may be worthwhile to include a mention in the discussion section of our manuscript which has been described in the previous comment/answer above. The added section on pages 9-10 reads as follows:

“While not investigated in the current study, foods that traditionally accompany eggs, including bacon and sausages, may be contributing to the increased sodium intake in the dietary pattern. This may imply the cooking methods involved with preparing and serving eggs and higher sodium foods that accompany eggs, may require additional scientific evaluation. Further investigation in this area is recommended, particularly with the release of a new report in older adults that showed higher consumption of cholesterol or eggs was associated with a small increase in CVD and all-cause mortality [30]. However, the study did not consider foods often consumed with eggs, including higher added sugar-, saturated fat-, and sodium-containing foods, like bacon, sausages, pancakes, waffles, and syrups.”

Author’s Addition: In addition to your comments, we have added a sentence in the conclusion to read as follows:

“The present study further illustrates an opportunity to communicate the benefits linked to egg consumption to individuals that influence children and adolescents, including parents, school nutrition organizations, educators, and to dietary guideline advisory committees globally.”

Reviewer 2 Report

The manuscript by Papanikolaou and Fulgoni III described the associations with egg consumption and nutrient intakes, diet quality and growth outcomes relative to non-egg consumers based on the survey of the U.S. National Health and Nutrition Examination Survey (NHANES) 2001-2012. The results showed that egg consumption was positively related with daily lutein + zeaxanthin and DHA intake. This is an interesting and well-designed study based on the previous studies from the authors. Overall, the manuscript was clearly written, and provided an updated investigation of egg consumption and nutrient intakes. For the benefit of the reader, there are, however, still several questions need to be answered and clarified. The following comments and suggestions should be taken into account in order to improve the overall quality and readability of the manuscript.

Minor concern:

1.      Please define the meaning of “egg consumption”  because there are quite a lot of foods containing eggs.

2.      Please raise a suggestion of nutrition education for children and adolescents, according to the results of this study.

Author Response

Dear Reviewer,

Thank you for taking the time to review our paper and provide feedback. In addition to checking spelling and grammar, we have provided answers and comments to your questions and suggestions below. Please let us know if you have any further questions or suggestions.

Sincerely,

Yanni Papanikolaou and Victor Fulgoni

Reviewer’s Question: Please define the meaning of ‘egg consumption’ because there are quite a lot of foods containing eggs.

Authors’ Response: Thank you for following up with us on this question. We agree that egg consumption can include many foods, and thus, we have revised our definition in Section 2.1 on page 3 to read as follows:

“Egg consumption was defined strictly as participants that consumed only eggs, poached eggs, scrambled eggs and omelets while excluding egg-containing mixed dishes (i.e., egg-containing sandwiches, breakfast burritos, and all egg-containing bakery foods, including cakes, breads, cookies, and biscuits).

Reviewer’s Suggestion: Please raise a suggestion of nutrition education for children and adolescents, according to the results of this study.

Author’s Response: Thank you for providing this recommendation. We agree with your suggestion and have added a sentence in the conclusion on page 10 to read as follows:

“The present study further illustrates an opportunity to communicate the benefits linked to egg consumption to individuals that influence children and adolescents, including parents, school nutrition organizations, educators, and to dietary guideline advisory committees globally.”
